# Methane-Oxidizing Communities in Lichen-Dominated Forested Tundra Are Composed Exclusively of High-Affinity USCα Methanotrophs

**DOI:** 10.3390/microorganisms8122047

**Published:** 2020-12-21

**Authors:** Svetlana E. Belova, Olga V. Danilova, Anastasia A. Ivanova, Alexander Y. Merkel, Svetlana N. Dedysh

**Affiliations:** Winogradsky Institute of Microbiology, Research Center of Biotechnology of the Russian Academy of Sciences, Leninsky Ave. 33/2, Moscow 119071, Russia; svet-bel@mail.ru (S.E.B.); vinnigo@gmail.com (O.V.D.); ivanovastasja@gmail.com (A.A.I.); alexandrmerkel@gmail.com (A.Y.M.)

**Keywords:** atmospheric methane oxidation, forested tundra, acidic soils, methanotrophic bacteria, USCα group, *pmoA* gene, bacterial diversity

## Abstract

Upland soils of tundra function as a constant sink for atmospheric CH_4_ but the identity of methane oxidizers in these soils remains poorly understood. Methane uptake rates of −0.4 to −0.6 mg CH_4_-C m^−2^ day^−1^ were determined by the static chamber method in a mildly acidic upland soil of the lichen-dominated forested tundra, North Siberia, Russia. The maximal CH_4_ oxidation activity was localized in an organic surface soil layer underlying the lichen cover. Molecular identification of methanotrophic bacteria based on retrieval of the *pmoA* gene revealed Upland Soil Cluster Alpha (USCα) as the only detectable methanotroph group. Quantification of these *pmoA* gene fragments by means of specific qPCR assay detected ~10^7^
*pmoA* gene copies g^−1^ dry soil. The *pmoA* diversity was represented by seven closely related phylotypes; the most abundant phylotype displayed 97.5% identity to *pmoA* of *Candidatus* Methyloaffinis lahnbergensis. Further analysis of prokaryote diversity in this soil did not reveal 16S rRNA gene fragments from well-studied methanotrophs of the order *Methylococcales* and the family *Methylocystaceae*. The largest group of reads (~4% of all bacterial 16S rRNA gene fragments) that could potentially belong to methanotrophs was classified as uncultivated *Beijerinckiaceae* bacteria. These reads displayed 96–100 and 95–98% sequence similarity to 16S rRNA gene of *Candidatus* Methyloaffinis lahnbergensis and “*Methylocapsa gorgona”* MG08, respectively, and were represented by eight species-level operational taxonomic units (OTUs), two of which were highly abundant. These identification results characterize subarctic upland soils, which are exposed to atmospheric methane concentrations only, as a unique habitat colonized mostly by USCα methanotrophs.

## 1. Introduction

Methane (CH_4_) is one of the most impactful greenhouse gases, which has contributed about 20% of the additional radiative forcing accumulated in the lower atmosphere since 1750 [1,2,3,4]. Emissions and concentrations of CH_4_ are continuing to increase; the current atmospheric CH_4_ mixing ratio is 1.8 ppmv. Major biological net sinks for atmospheric methane are well aerated upland soils [5,6,7]. Atmospheric CH_4_ uptake in these soils is due to activity of aerobic methanotrophic bacteria, which utilize methane as a source of energy [8,9,10,11]. Aerobic methanotrophs comprise a number of bacterial genera within the *Gamma*- and *Alphaproteobacteria*, as well as the *Verrucomicrobia* [12]. A key enzyme of the methanotrophic metabolism is particulate methane monooxygenase (pMMO), which is present in most currently described methanotroph species. Accordingly, the *pmoA* gene coding for the active-site polypeptide of pMMO is the most frequently used molecular marker in cultivation-independent detection of aerobic methanotrophs [13]. 

The apparent affinity for CH_4_ measured for pure cultures of most currently described methanotroph species is in the range of 1–10 µm, while the respective values determined for upland soils are in the nanomolar range [14]. Microorganisms that are able to oxidize atmospheric CH_4_, therefore, are often referred to as “high-affinity” methanotrophs. The identity of high-affinity methane oxidizers in terrestrial ecosystems has been the focus of considerable microbiological research. The first evidence that as-yet-uncultivated methanotroph group is involved in atmospheric CH_4_ oxidation in upland soils was obtained by both labeling phospholipid fatty acids of methanotrophs with ^14^CH_4_ and analyzing the *pmoA* gene library [15]. The *pmoA* sequences retrieved in this study could not be affiliated to any of the earlier described methanotrophs and displayed only a distant relationship to *pmoA* gene from *Methylocapsa acidiphila*, an alphaproteobacterium isolated from acidic peat [16,17]. This previously unknown group of *pmoA* sequences, called Upland Soil Cluster *Alphaproteobacteria* (USCα) after Knief et al. [18], has been recovered from many acidic and pH-neutral boreal upland soils [18,19,20,21,22,23]. These bacteria were also identified as the predominant methanotroph group in acidic carbon-poor cryosols at Axel Heiberg Island in the Canadian high Arctic, which were shown to consistently consume atmospheric methane [24].

The exact phylogenetic position of USCα methanotrophs remained unknown for a long time but their tentative affiliation to the alphaproteobacterial family *Beijerinckiaceae* was suggested by several metagenomic studies [25,26,27]. Reconstruction of a draft genome of USCα methanotroph enabled the first insights into the metabolic potential and environmental adaptation strategies of these methanotrophs [27]. The use of multilocus sequence analysis suggested placement of the metagenomic assembly obtained in the latter study in the new, *Methylocapsa*-related genus of the family *Beijerinckiaceae*, *Candidatus* Methyloaffinis lahnbergensis.

Recent isolation of a pure culture of the first cultivated member of the USCα clade, strain MG08, provided valuable insights into the physiology and metabolism of these methanotrophs [28]. CH_4_ oxidation experiments and ^13^C-single cell isotope analyses proved the ability of strain MG08 to grow at atmospheric concentrations of CH_4_ and to assimilate carbon from both CH_4_ and CO_2_. The genome of strain MG08 encodes the serine cycle for assimilation of carbon from CH_4_ and CO_2_, and CO_2_ fixation through the reductive glycine pathway. This strain also fixes N_2_ and expresses the genes for a high-affinity hydrogenase and carbon monoxide dehydrogenase, suggesting that oxidation of the atmospheric trace gases, CO and H_2_, provides additional energy sources [28]. Based on the results of comparative genome analysis, strain MG08 was classified as a member of the genus *Methylocapsa* and was tentatively named “*Methylocapsa gorgona*” MG08. Although the PmoA sequence of *Methylocapsa gorgona* MG08 clusters within the radiation of USCα PmoA clade, it does not correspond to the major group of PmoA sequences, which are most often retrieved from acidic upland soils. The exact diversity range of USCα methanotrophs, therefore, remains unclear. 

This study was undertaken in order to extend the currently available information about the phylogenetic diversity of USCα methanotrophs using unique samples of Russian tundra upland soil, where USCα represents the only detectable methanotroph group.

## 2. Materials and Methods 

### 2.1. Study Site 

Field studies were carried out in July 2014 in the lichen-dominated pine (*Pinus sibirica*) forest within a permafrost-free zone of tundra near the Nadym town, the Yamalo-Nenetsky Autonomy, North Siberia (65°36′07.1′′ N, 72°44′ 39.5′′ E) (Figure 1a). The soil vegetation cover was composed of *Cladonia* and *Cetraria* species; *Vaccinium* spp., *Ledum palustre*, and *Polytrichum commune* were also present (Figure 1b,c). Analysis of the soil profile revealed the presence of litter, thin organic layer, gray sand, and sandy subsoil (Figure 1d). A set of individual soil samples (~500 g each) was collected over the soil profiles of three sampling plots located on a distance of 20–30 m from each other. The collected soil was transported to the laboratory, homogenized, and frozen for further molecular analyses within 1 day after sampling.

### 2.2. Methane Flux Measurements 

Methane fluxes were measured by the static chamber method [29,30]. Concentration of CH_4_ in the samples was determined using a Kristall-5000 chromatograph (Khromatek, Yoshkar-Ola, Russia) with a flame ionization detector. Methane flux was calculated in mg CH_4_-C m^−2^ day^−1^.

### 2.3. Determination of CH_4_ Oxidation Activity of Soil Samples 

Methane oxidation rates were determined as described in our previous study [31]. Briefly, weighted portions of wet soil (10 g) sampled from different layers were placed in sterile glass vials 160 mL in volume, which were then sealed hermetically. Methane was injected in the flasks up to the concentration of about 1000 ppm. The vials were incubated at room temperature for 48 h. Samples of the gas phase (0.5 mL) were taken from the flasks periodically and analyzed for methane concentration on a Kristall 5000 chromatograph (Khromatek, Russia). The rate of methane oxidation by the soil samples was calculated in μg CH_4_ g^−1^ of wet soil h^−1^.

### 2.4. Chemical Analysis of the Soil Samples

The analysis was carried out by certified standard techniques in the analytical center of the MSU Soil Science faculty. The total contents of organic C and N in the soil were measured using the vario MACRO Cube analyzer (Elementar Analysensysteme GmbH, Germany). Sulfate concentrations were measured using the Dionex ICS-2000 Ion Chromatography System (Dionex, Sunnyvale, CA, USA). The contents of Ca, Mg, P, and Fe were determined by means of inductively coupled plasma mass spectrometry (ICP-MS Agilent 7500a, Agilent, Santa Clara, CA, USA).

### 2.5. DNA Extraction, PCR Amplification, and Illumina Sequencing 

Since highest methane oxidation rates were detected in soil collected from the surface organic layer at a depth of 2–5 cm (see Results), these samples were used for molecular diversity studies. Total DNA extracts were obtained from three individual soil samples (~0.5 g wet weight) collected from the three sampling plots. The procedure used for DNA extraction and purification is described in our earlier published study [32]. Three DNA extracts were obtained from each soil sample and pooled together for further use in PCR. Fragments of 16S rRNA gene corresponding to the V4 region were PCR-amplified from the DNA samples, purified, and sequenced using the 300PE protocol on MiSeq System (Illumina, USA) as outlined elsewhere [32]. The obtained reads were quality filtered and trimmed using CLC Genomics Workbench 7.5 (Qiagen, Germany), after which overlapping paired-end reads were merged with SeqPrep tool (https://github.com/jstjohn/SeqPrep).

### 2.6. Bioinformatic Analysis of Amplicon Sequences 

The 16S rRNA gene reads obtained from the studied soil samples were analyzed with QIIME 2 v.2018.8 (https://qiime2.org) [33]. Sequence quality control, denoising, and chimera filtering were performed by using DADA2 plugin [34]. The reads were clustered into operational taxonomic units (OTUs) by using VSEARCH plugin [35] with open-reference function and Silva v. 132 database [36,37] with 97% identity.

### 2.7. Molecular Identification of Methanotrophic Bacteria

Total DNA extracted from the soil was used as a template in PCR with the primers A189f (GGNGACTGGGACTTCTTG) and A682r (GAASGCNGAGAAGAASGC) specific to the *pmoA* gene coding for the active-site polypeptide of particulate methane monooxygenase (pMMO) [38]. DNA from the methanotrophic bacterium *Methylocystis heyeri* H2^T^ was used as the positive control. PCR was performed in a PE GeneAmp PCR System 9700 (Perkin-Elmer Applied Biosystems, United States) thermocycler. The reaction mixture contained 0.5–1.0 μL DNA, A189f and A682r primers, 1 μL each, 50 μL MasterMix (Promega), and sterile water to the total volume of 100 μL. The thermal profile of the reaction was as follows: initial denaturation (1 min at 94 °C); 33 cycles of denaturation (1 min at 94 °C), primer annealing (1 min at 55 °C), and elongation (1 min at 72 °C), followed by final elongation step (7 min at 72 °C). PCR products were examined by electrophoresis in 1.2% agarose gel followed by ethidium bromide staining and visualization of reaction products under a UV transilluminator. Products of three independent reactions obtained with the template DNA of three extracts were pooled together and used for further cloning. Cloning was performed using the pGem-T Easy Vector System II (Promega) according to the manufacturer’s recommendations. Recombinant clones were selected using the blue-white screening and screened for the correct insert size (approximately 530 bp) with vector-specific primers T7 and Sp6 (Promega). Plasmid DNA was purified using a Wizard Plus Minipreps DNA Purification System (Promega). Nucleotide sequences of the cloned DNA fragments were determined on an ABI 377A (Perkin-Elmer Applied Biosystems) sequencer. Phylogenetic trees were constructed using the ARB software package [39].

### 2.8. qPCR-based Quantification of Methanotrophs 

Quantification of *pmoA* genes was performed according to a previously described method [40] on a the StepOnePlus™ Real-Time PCR System (Thermo Fisher Scientific, USA) using qPCRmix-HS SYBR Kit (Evrogen, Russia). Concentration of *pmoA* genes was estimated using the primer system highly specific to the *pmoA* sequences retrieved from the examined soil samples at the previous stage. A primer system was designed using Primrose software [41]: pmoA-Cl-F (5′-GCA ATA TGG CAC GCT GAT GT-3′) and pmoA-Cl-R (5′-ATG TAT TCG GGC ATC GAG GTA-3′). qPCR thermal program was adjusted experimentally according to the standard protocol: 94 °C—20”; 62 °C—20”; 72 °C—15”; primers conc.—0.5 μM; Mg++ conc.—3 mM. One of the *pmoA* amplicons inserted in the pGEM^®^-T vector was used to generate a standard curve. Standards, samples and negative controls were run in triplicate. The correlation coefficient for standard curve was 0.997.

### 2.9. Sequence Accession Numbers 

The 16S rRNA gene reads retrieved using Illumina pair-end sequencing from the soil (raw data) have been deposited under the BioProject number PRJNA344855 in the NCBI Sequence Read Archive, with the accession numbers SRX2207619-SRX2207621. Nucleotide sequences of the *pmoA* gene fragments determined in this study were deposited in GenBank under accession nos. KX534007–KX534056. Identical nucleotide sequences were not deposited.

## 3. Results and Discussion

### 3.1. Measurements of CH_4_ Fluxes

Measurements of *in situ* surface CH_4_ fluxes at the studied forested tundra site in July 2014 showed atmospheric CH_4_ uptake with rates −0.4 to −0.6 mg CH_4_-C m^−2^ day^−1^. These CH_4_ uptake rates are comparable with those measured at the McGill Arctic Research Station at Expedition Fjord, AHI, Canada (−0.1 to −0.8 mg CH_4_-C m^−2^ day^−1^) [24], Quttinirpaaq National Park, Canada (−1.37 ± 0.06 mg CH_4_ m^−2^ day^−1^) [42], and other terrestrial systems at lower latitudes (−0.1 to −1.0 mg CH_4_-C m^−2^ day^−1^) [43].

### 3.2. Distribution of Methane Oxidation Activity over the Soil Profile

Soil profiles in the three sampling plots were highly similar to each other and included litter, thin dark-colored organic layer, gray sand, and sandy subsoil (see 1–4 in Figure 1d). The surface organic layer and the underlying sandy soil displayed a number of contrasting characteristics (Table 1). The former had a pH 4.1, while the latter was nearly neutral (pH 6.1). The content of organic carbon in the surface dark-colored layer was one order of magnitude higher than that in the sandy soil (Table 1). The maximal methane oxidation rates, 0.15–0.18 µg CH_4_ g^−1^ h^−1^, were detected in samples taken from the organic soil layer, while CH_4_ oxidation activity in the sandy soil layer was nearly non-measurable (Figure 1e). Further molecular analyses, therefore, were performed with DNA extracts obtained from the surface organic soil layer.

### 3.3. Identification and Quantification of Methanotrophs Using pmoA-Based Analysis

A total of 222 *pmoA* gene clone sequences were retrieved from the studied DNA extracts. Of these, only 50 nucleotide sequences were unique and were deposited in the GenBank. These 50 sequences represented 7 closely related *pmoA* phylotypes grouped on the basis of 98% nucleotide sequence identity (Table 2).

All of them belonged to Beijerinckiaceae methanotrophs and affiliated with the USCα clade (Figure 2). The most abundant phylotype, *pmoA*-1, was represented by 148 sequences and displayed 97.5% nucleotide identity to *pmoA* of *Candidatus* Methyloaffinis lahnbergensis. If grouped on the basis of 93% nucleotide sequence identity as recommended by Degelmann et al. (2010) [21], all *pmoA* sequences obtained in our study belong to one species-level OTU, thus suggesting extremely low methanotroph diversity in the tundra soil.

The pool of *pmoA* sequences retrieved in this study was used to design the primers for specific detection of the target methanotroph group (see Methods). The qPCR-detected *pmoA* gene copy numbers were highly similar in all three sampling plots and were within a range of 5.9–8.3 × 10^6^ copies g^−1^ of wet soil or 6.6–9.7 × 10^6^ copies g^−1^ of dry soil (given water content of 10.7–14.7%). These values fall within the range of USCα *pmoA* gene numbers reported earlier for various soils exhibiting atmospheric CH_4_ consumption, 0.9 × 10^6^–1.2 × 10^8^ copies per gram of dry soil [7,21,44,45]. Highest USCα *pmoA* gene numbers, 0.3–1.2 × 10^8^ copies per gram of dry soil, have so far been reported for boreal forest soils in Germany [21].

### 3.4. Molecular Analysis of Bacterial Diversity in Tundra Soil

A total of 712,393 partial 16S rRNA gene sequences (mean amplicon length 253 bp) were retrieved from the examined tundra soil. Of these, 496,695 reads were retained after quality filtering, denoising, and removing chimeras. Predominant prokaryote groups in the communities from three sites were the *Acidobacteria* (27.5 ± 0.1% of the total number of 16S rRNA gene fragments, mean ± SE), *Alphaproteobacteria* (25.2 ± 1.0%), *Planctomycetes* (12.3 ± 0.8%), *Actinobacteria* (12.1 ± 0.9%), *Verrucomicrobia* (9.3 ± 0.1%), G*ammaproteobacteria* (5.3 ± 0.7%), and *Deltaproteobacteria* (3.4 ± 0.2%) (Figure 3A). Minor bacterial groups included WPS-2*, Chloroflexi*, *Cyanobacteria*, *Firmicutes*, and others.

The largest group among the *Alphaproteobacteria* was the order *Rhizobiales* (64 ± 0.7% of the number of 16S rRNA gene fragments affiliated with the *Alphaproteobacteria***)**. Two groups of *Rhizobiales*-affiliated sequences were represented by reads taxonomically classified within the genera *Roseiarcus* (20.8 ± 1.9% of all *Rhizobiales*-affiliated reads) and *Bradyrhizobium* (20.9 ± 4.1%) (Figure 3B).

The third group of *Rhizobiales*-affiliated sequences belonged to uncultured members of the *Xanthobacteraceae* (29.2 ± 2.5%). Finally, the fourth group of sequences (25.8 ± 2.8%) was identified as belonging to uncultured *Beijerinckiaceae*. Further analysis of these sequences revealed that they are represented by 8 phylotypes (Table 3).

Two of these phylotypes, 16S-1 and 16S-2, were most abundant and were represented by 11,712 and 9446 reads, comprising 2.4 and 1.9% of all bacterial sequences retrieved from the studied soil. Phylotypes 16S-1 and 16S-2 displayed 99 and 100% sequence similarity to 16S rRNA gene sequence of *Candidatus* Methyloaffinis lahnbergensis and 98.6 and 97.6% sequence similarity to 16S rRNA gene sequence of “*Methylocapsa gorgona”* MG08, respectively (Table 3, Figure 4).

Notably, no 16S rRNA gene fragments from well-studied methanotrophs of the order *Methylococcales* and the family *Methylocystaceae* were detected within the pool of reads retrieved in our study, which fully corresponds to the results of *pmoA*-based diversity analysis.

The results obtained in our study, therefore, characterize subarctic sandy upland soils as a unique habitat colonized mostly or exclusively by USCα methanotrophs. As shown by the recent 16S rRNA gene-based environmental distribution survey of USCα methanotrophs [27], arctic and subarctic soils are among the major habitats of these bacteria. The ecological advantage of USCα members over other methanotrophs, most likely, is explained by the fact that these soils, in contrast to arctic wetlands and boreal upland soils, are exposed to atmospheric methane concentrations only. While arctic wetlands may emit methane, arctic desert soils function as a constant sink for atmospheric CH_4_ [42]. One of our previous studies on methanotroph diversity was performed in a subarctic wetland with mosaic cover of *Sphagnum* mosses and lichens [31]. This wetland is in a close proximity to the study site examined in this work (a distance of 14.5 km). While USCα methanotrophs were also detected in that subarctic wetland, the indigenous methanotroph community was by far more diverse and included conventional type I and type II methanotrophs as well [31]. Interestingly, members of the genera *Methylocystis* and *Methylosinus*, which contain Pmo2 enzyme catalyzing oxidation of CH_4_ at atmospheric levels [46,47,48] and are commonly detected in upland boreal soils [5,7,18,21], were absent from the studied tundra soil. As suggested earlier, these methanotrophs cannot sustain activity and growth on atmospheric methane alone, and their long-term survival depends on availability of methane produced in anaerobic micro-sites of upland soils [5]. Such an *in situ* methane production does not occur in well-aerated sandy soils examined in this study and, therefore, Pmo2-possessing *Methylocystis* and *Methylosinus* species cannot compete with USCα methanotrophs for colonizing this niche.

As seen from Figure 2, most methanotrophs detected in lichen-dominated tundra soil were closely related to *Candidatus* Methyloaffinis lahnbergensis [27] and not to “*Methylocapsa gorgona*” MG08 [28]. Although phylogenetically close, these methanotrophs may possess some differences in physiology and adaptation potential, which are reflected in their distribution pattern. Isolation of *Candidatus* Methyloaffinis-like bacteria remains one of the key challenges in methanotroph cultivation and subarctic sandy soils are one of the promising sources for these isolation studies.

In summary, our study showed that upland soils of a lichen-dominated forested tundra function as a sink for atmospheric CH_4_. Methane-oxidizing communities in these soils are composed exclusively of high-affinity USCα methanotrophs, which are localized in a very thin organic surface soil layer underlying the lichen cover. Possible disturbances of this surface soil layer due to anthropogenic activities may result in degradation of this fragile methane-oxidizing microbial filter and loss of its function.

## Figures and Tables

**Figure 1 microorganisms-08-02047-f001:**
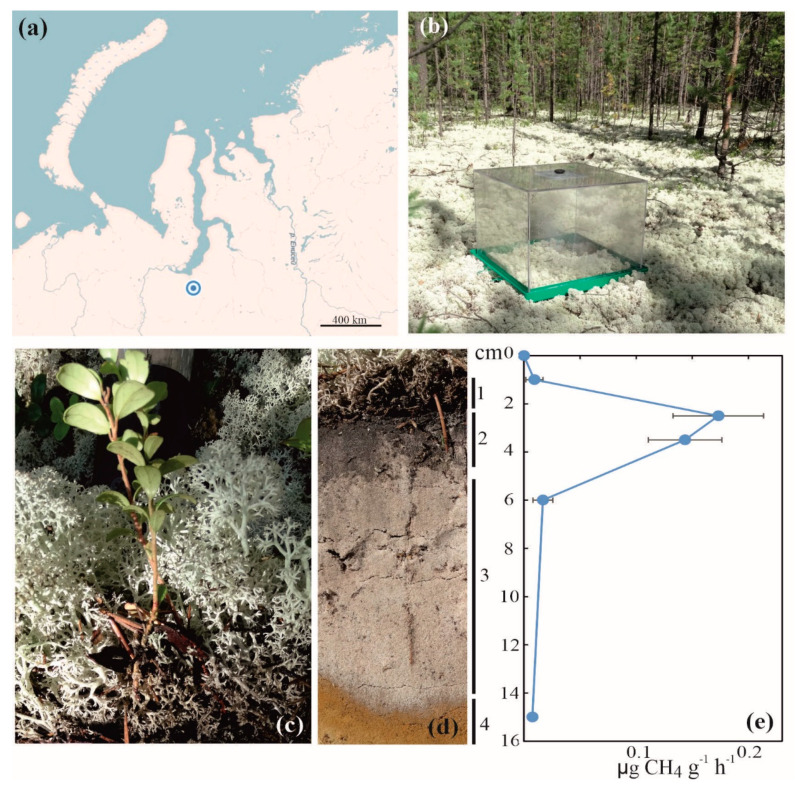
Location and some characteristics of the study site: (**a**) location of the study site on the map of northern Russia; (**b**) CH_4_ flux measurements by static chambers; (**c**) vegetation cover at the study site; (**d**) soil profile: 1—litter, 2—organic layer, 3—gray sand, and 4—sandy subsoil; (**e**) distribution of CH_4_ oxidation activity over the soil profile.

**Figure 2 microorganisms-08-02047-f002:**
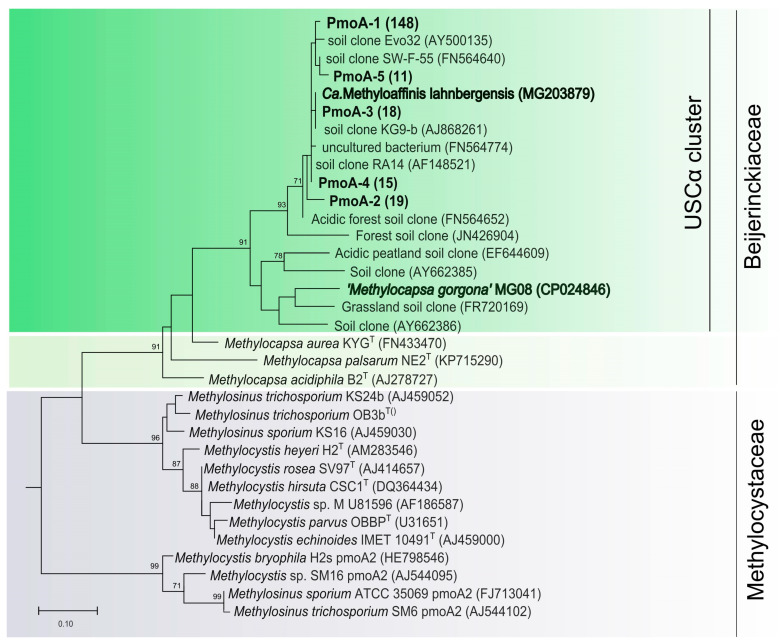
Phylogenetic tree constructed based on 201 amino acid sites of partial PmoA sequences obtained from a forested tundra soil (shown in bold), as well as PmoA from representative members of the families *Beijerinckiaceae* and *Methylocystaceae* and some environmental clone sequences. The scale bar corresponds to 0.1 substitutions per amino acid position.

**Figure 3 microorganisms-08-02047-f003:**
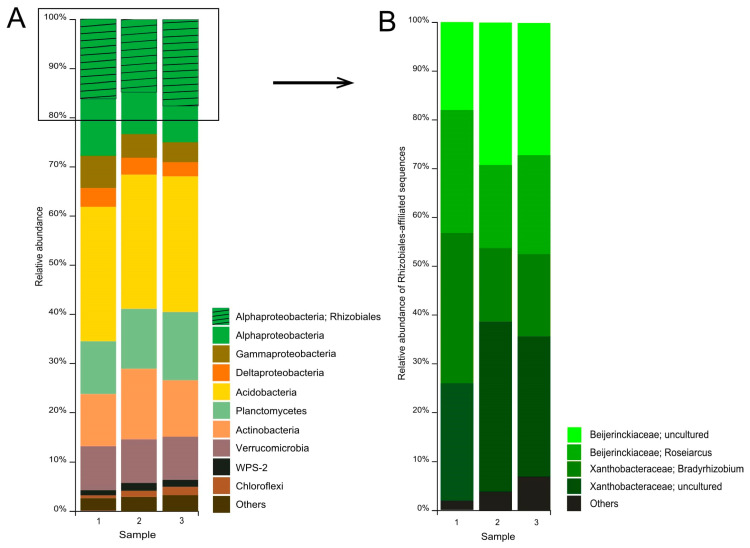
Bacteria community composition in forested tundra soil according to the results of Illumina-based sequencing of 16S rRNA genes. The composition is displayed at the (**A**) phylum level and (**B**) family level of the alphaproteobacterial order *Rhizobiales*.

**Figure 4 microorganisms-08-02047-f004:**
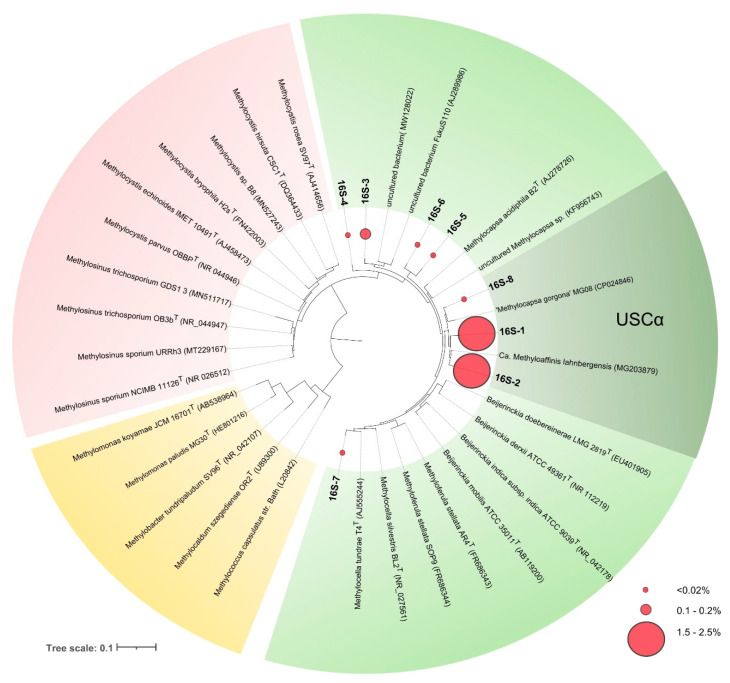
16S rRNA gene-based maximum-likelihood tree showing the phylogenetic position of sequences retrieved from acidic tundra soil in relation to *Candidatus* Methyloaffinis lahnbergensis, “*Methylocapsa gorgon*a” MG08 and some representatives of the family *Beijerinckiaceae* (shown in green), *Methylocystaceae* (red), and *Methylococcaceae* (yellow). Red circles of different sizes show relative abundances of the corresponding OTUs. The scale bar corresponds to 0.1 substitutions per nucleotide position.

**Table 1 microorganisms-08-02047-t001:** Chemical composition of soil samples.

Parameter		Organic Layer	Gray Sand
C org	%	1.51 ± 0.30	0.11 ± 0.02
N total		0.06 ± 0.01	0.01 ± 0.002
N-NO_3_	mg/kg	7.30 ± 0.55	7.00 ± 0.53
N-NH_4_		14.00 ± 1.05	9.00 ± 0.68
P total		165.0 ± 33.0	39.0 ± 7.8
SO_4_^2-^		19.40 ± 4.85	1.50 ± 0.38
Ca		13600 ± 2176	440.0 ± 70.4
Mg		450.0 ± 72.0	26.0 ± 4.16
Fe		1821.0 ± 291.0	1621.0 ± 259.4
pH		4.1 ± 0.1	6.1 ± 0.1

**Table 2 microorganisms-08-02047-t002:** The phylotypes of *pmoA* gene sequences retrieved from lichen-dominated tundra soil.

Phylotype	Number of Sequences	GenBank Accession Numbers	Identity to *Cand*. M. lahnbergensis (%)	Identity to *M. gorgona* (%)
*pmoA*-1 *	148	KX534008, KX534009, KX534010, KX534011, KX534012, KX534013, KX534014, KX534015, KX534016, KX534017, KX534018, KX534020, KX534021, KX534022, KX534023, KX534025, KX534026, KX534027, KX534028, KX534029, KX534031, KX534035, KX534037, KX534038, KX534039, KX534040, KX534041, KX534044, KX534046, KX534048, KX534051, KX534052, KX534055, KX534056	97.5	84.2
*pmoA*-2 *	19	KX534053, KX534045, KX534030, KX534047, KX534036, KX534033	96.4	83.1
*pmoA*-3 *	18	KX534034, KX534049, KX534019	99.2	83.5
*pmoA*-4 *	15	KX534032, KX534042	95.8	84.0
*pmoA*-5 *	11	KX534043	95.0	80.8
*pmoA*-6	8	KX534007, KX534054	96.4	83.1
*pmoA*-7	3	KX534050	97.5	83.8

* The phylotypes included in the phylogenetic tree (see Figure 2).

**Table 3 microorganisms-08-02047-t003:** Major operational taxonomic units (OTUs) of 16S rRNA gene sequences affiliated with the family *Beijerinckiaceae*.

OTU ID	Relative Abundance (%)	Taxonomy	Closest Silva Match (Similarity, %)	Reported Habitat	Similarity MG203879 (%)	Similarity M*. gorgona* (%)
16S-1	2.358	*Beijerinckiaceae* uncultured	AY913480 (100)	bulk soil	99	98.6
16S-2	1.902	*Beijerinckiaceae* uncultured	AY913598 (100)	limestone cave	100	97.6
16S-3	0.167	*Beijerinckiaceae*	KX509291 (100)	rainwater	98.6	98.1
16S-4	0.021	*Beijerinckiaceae*	KT182565 (100)	bioreactor	96.6	96.2
16S-5	0.019	*Beijerinckiaceae*	JQ905994 (99)	acid mine	96.6	97.1
16S-6	0.016	*Beijerinckiaceae*	FPLS01031837 (100)	unknown	98.6	98.1
16S-7	0.01	*Beijerinckiaceae*	KF100807 (99)	human skin	96.6	95.7
16S-8	0.004	*Beijerinckiaceae* uncultured	HG528987 (100)	peatland	98.1	98.6

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
