# Peer review of "Methane-Oxidizing Communities in Lichen-Dominated Forested Tundra Are Composed Exclusively of High-Affinity USCα Methanotrophs"

_microorganisms, 2020, doi:10.3390/microorganisms8122047_

Round 1

Reviewer 1 Report

The study from Belova et al. describe the molecular identification of methanotrophs belong to the USCα cluster that are responsible for atmospheric methane oxidation in a tundra soil.

The study from Belova et al is mostly descriptive, but the findings are relevant to better understand the ecology and phylogeny of methanotrophs involved into atmospheric methane fixation. The study would benefit from improving the abstract with more context and drawing a conclusion that would explain the implication of the results beyond this particular stidy in order to see the bigger picture.

Specific comments

Abstract : There are not context in the abstract, the study starts directly with some results. The abstract would benefit for a contextualisation about why it is important to better characterize methanotrophs diversity and what are the implication for future climate change studies.

Line 36. What do the author mean by ‘long-liver greenhouse gases’ ? please specify.

Line 38. What do the author mean by ‘comparatively’ here ? this sentence is not clear.

Line 82. Why is PmoA not written pmoA and italicised here ?

Figure 1.a. There are no scale on the map

Figure 3. That seems unnecessary to repeat ‘Bacteria’ and ‘Rhizobiales’ at the beginning of the legend for each graph, especially that phylum and family level are already stated in the legend. Also, in Figure 3B, the y-axis refers the relative abundance, while only the Rhizobiales were shown, so the axis should be corrected to Relative abundance of Rhizobiales, or the Y-axis should be corrected to reflect the real relative abundance compared to the rest of the microbial community.

Author Response

Comment: Abstract: There are not context in the abstract, the study starts directly with some results. The abstract would benefit for a contextualisation about why it is important to better characterize methanotrophs diversity and what are the implication for future climate change studies.

Response: Ok, we have added one introductory sentence.

Comment: Line 36. What do the author mean by ‘long-liver greenhouse gases’ ? please specify.

Response: ‘Long-lived greenhouse gases’ are those gases, which are sufficiently mixed throughout the troposphere because their atmospheric lifetimes are much greater than the time scale of a few years for atmospheric mixing. This is a common term in atmospheric chemistry. This term was omitted from the revised manuscript because we were asked to replace this sentence anyway.

Comment: Line 38. What do the author mean by ‘comparatively’ here ? this sentence is not clear.

Response: We have corrected this sentence in order to make it clear to the reader and used the wording ‘well aerated soils’.

Comment: Line 82. Why is PmoA not written pmoA and italicised here ?

Response: We’re discussing amino acid identity in this sentence. This is a standard way to write names of proteins.

Comment: Figure 1.a. There are no scale on the map

Response: Ok, the scale is included now.

Comment: Figure 3. That seems unnecessary to repeat ‘Bacteria’ and ‘Rhizobiales’ at the beginning of the legend for each graph, especially that phylum and family level are already stated in the legend. Also, in Figure 3B, the y-axis refers the relative abundance, while only the Rhizobiales were shown, so the axis should be corrected to Relative abundance of Rhizobiales, or the Y-axis should be corrected to reflect the real relative abundance compared to the rest of the microbial community.

Response: this figure has been replaced with its corrected version.

Reviewer 2 Report

Belova et al. studied the Methane-oxidizing communities in lichen-dominated forested tundra are composed exclusively of high-affinity USCα methanotrophs. In my opinion this work describes an important topic about the Upland Soil Cluster Alpha (USCα) methanotrophs that have been identified in subarctic upland soils, which are exposed to atmospheric methane concentrations and classified as uncultivated Beijerinckiaceae. Generally, this study is interesting and meaningful, and the manuscript is well written and fully discussed. It can be accepted after minor revision.

  1. Figure 1.e: Please change the unit record, and please use a dot instead of a comma as the decimal separator.
  2. Materials and Methods: “Methane flux measurements” and “Determination of CH4 oxidation activity of soil samples” – this is the same description as in previous study, please cite earlier paper.
  3. In the manuscript, the reviewer did not find detail information regarding the number of the samples. Three plots were selected, but there is no information on the number of soil samples. Were soil samples taken at each level?
  4. How many replicates of total DNA isolation used for molecular diversity studies were performed?
  5. I have doubts about BioProject and SRA numbers. In the PRJNA344855 I see only 2 BioSamples, while the authors analyze the results from 3 samples. On the other hand, in the database are 6 different SRA numbers assigned to this BioProject, but the authors only mention one. Please verify.
  6. Table 3. The title of the table should not be italicized.
  7. Please justify the Results and Discussion part.
  8. I recommend adding a summary of the work or adding a conclusions point.
  9. Please read the authors' guide carefully and correct references.

Author Response

Comment:  Figure 1.e: Please change the unit record, and please use a dot instead of a comma as the decimal separator.

Response: corrected as recommended.

Comment: Materials and Methods: “Methane flux measurements” and “Determination of CH4 oxidation activity of soil samples” – this is the same description as in previous study, please cite earlier paper.

Response: We have cited the study of Danilova et al. 2016 with regard to measurements of methane oxidation. The methodology of methane flux measurements is described in references 29, 30.

Comment:  In the manuscript, the reviewer did not find detail information regarding the number of the samples. Three plots were selected, but there is no information on the number of soil samples. Were soil samples taken at each level?

Response: The samples were collected from all soil layers (over the soil profile) as stated in lines 90-91. All of these samples were used to measure methane oxidation activity (as shown in Fig 1e). Molecular studies, however, were performed only with the soil samples displaying highest methane oxidation activity, i.e. with samples collected from the surface organic layer at a depth of 2-5 cm. This is now explained in line 120-122.

Comment: How many replicates of total DNA isolation used for molecular diversity studies were performed?

Response: These details are now given in lines 122-125.

Comment: I have doubts about BioProject and SRA numbers. In the PRJNA344855 I see only 2 BioSamples, while the authors analyze the results from 3 samples. On the other hand, in the database are 6 different SRA numbers assigned to this BioProject, but the authors only mention one. Please verify.

Response: Sorry, this is our mistake. We provide three numbers now (SRX2207619-SRX2207621).

Comment: Table 3. The title of the table should not be italicized.

Response: corrected.

Comment: Please justify the Results and Discussion part.

Response: Combining Results and Discussion gave us a possibility to compare the results of our study with those obtained by other groups. This was especially convenient for comparing our values of methane fluxes and pmoA gene copy numbers to those in previously published studies. We believe that the reader will benefit from this way of presenting/comparing results.

Comment: I recommend adding a summary of the work or adding a conclusions point.

Response: Good suggestion. Done as recommended.

Comment: Please read the authors' guide carefully and correct references.

Response: The reference list has been corrected. Sorry for this mistake.